# Association between abdominal obesity and diabetic retinopathy in patients with diabetes mellitus: A systematic review and meta-analysis

**Shouqiang Fu**[1], **Liwei Zhang**[1], **Jing Xu**[2], **Ximing Liu**[1☯*], **Xiaoyun Zhu**[1☯*]

**1** Department of Endocrinology, Guang'anmen Hospital, China Academy of Chinese Medical Sciences, Beijing, China, **2** Department of Encephalopathy, Dongzhimen Hospital, Beijing University of Chinese Medicine, Beijing, China

☯ These authors contributed equally to this work.
* lxmhospital@126.com (XL); qiebenben@163.com (XZ)

## Abstract

### Objective

Previous studies have reported different opinions regarding the association between abdominal obesity and diabetic retinopathy (DR) in patients with diabetes mellitus (DM). In this study, we aimed to investigate this problem through a systematic review and meta-analysis to provide a basis for clinical interventions.

### Methods

A comprehensive search was conducted in the PubMed, Embase, and Web of Science databases up to May 1, 2022, for all eligible observational studies. Standardized mean differences (SMD) and 95% confidence intervals (CI) were evaluated using a random-effects model in the Stata software. We then conducted, publication bias assessment, heterogeneity, subgroup and sensitivity analyses.

### Results

A total of 5596 DR patients and 17907 non-DR patients were included from 24 studies. The results of the meta-analysis of abdominal obesity parameters showed statistically significant differences between DR and non-DR patients in both type 1 and type 2 diabetes. Waist circumference (WC) was higher in patients with DR than in the non-DR patients. In the waist-hip ratio (WHR) subgroup, the level of WHR was higher in patients with DR than that in non-DR patients. The association between abdominal obesity and mild to moderate nonproliferative DR or vision-threatening DR groups did not show any statistical difference. Subgroup analysis according to ethnicity showed that Caucasians had higher levels of combined abdominal obesity parameters than Asians.

**Data Availability Statement:** All relevant data are within the paper and its Supporting Information files.

**Funding:** The authors received no specific funding for this work.

**Competing interests:** The authors have declared that no competing interests exist.

## Conclusion

We found that abdominal obesity measured by WC and WHR is associated with DR in patients with type 1 and type 2 diabetes. This association is stronger in Caucasians than in Asians, where isolated abdominal obesity might be more related to DR. However, no correlation was found between abdominal obesity and varying degrees of diabetic retinopathy. Further prospective cohort studies with larger sample sizes are yet to be conducted to clarify our findings.

## Introduction

Diabetic retinopathy (DR) is the most common microvascular complication of diabetes and is one of the leading causes of blindness in the working-age population, worldwide [1]. Currently recognized risk factors affecting the development of DR include the duration of diabetes, elevated HbA1c, blood glucose, blood pressure, serum cholesterol, and low-density lipoprotein levels [2]. As one of the essential components of metabolic syndrome (MS), abdominal obesity, also known as central obesity, is an essential component of MS and is characterized by excessive accumulation of abdominal visceral fat. It is a crucial risk factor for metabolic diseases, such as diabetes and coronary heart disease, leading to insulin resistance and adipose tissue inflammation [3]. Meanwhile, conflicting results have been reported in recent clinical studies on the association between abdominal obesity and DR. Some studies have claimed that abdominal obesity is more associated with DR among people with diabetes mellitus (DM) than general obesity, as measured by body mass index (BMI) [4]. A study published in 2022 [5] announced that abdominal obesity was not associated with DR in patients with DM.

A previous meta-analysis [6] evaluating the association between abdominal obesity and DR among the Chinese population revealed that abdominal obesity defined by waist circumference (WC) is associated with the risk of DR, while the waist-hip ratio (WHR) is not. However, they did not compare the results among different ethnicities. Additionally, their analysis did not explore whether DR of different severities has an equal association with abdominal obesity. In addition, several recent clinical studies have used new measurement parameters to define abdominal obesity, such as lipid accumulation product (LAP) [4], Visceral fat area (VFA) [7] and visceral adiposity index (VAI) [5]. Currently, no meta-analysis have incorporated these parameters to investigate the association between abdominal obesity and DR. Therefore, given the paucity of evidence and limitations of previous studies, we carried out this updated meta-analysis to further evaluate the association between abdominal obesity parameters (WC, WHR, LAP, VFA, and VAI) and DR. Furthermore, we assessed the potential effects of different patient ethnicities, DR severity, and types of diabetes on the outcomes.

## Methods

We conducted this meta-analysis following the MOOSE (Meta-analyses of Observational Studies in Epidemiology) guidelines [8]. This study was registered in the INPLASY (ID: INPLASY202250091).

### Measurements and definitions

WC data were obtained directly from anthropometric measurements. The WHR and waist-height ratio (WHtR) were calculated by dividing WC by hip circumference or height. VFA

was measured using umbilicus level Computed Tomography (CT), Magnetic Resonance Imaging (MRI), or the bio-electrical impedance method. The calculation of LAP and VAI included triglyceride (TG) and/or high-density lipoprotein (HDL) values from the patient's fasting serum test [9].

DR was diagnosed using digital color fundus photography after pupil dilation in both eyes. Referring primarily to the Early Treatment for Diabetic Retinopathy Study (ETDRS) standards [10], DR was classified according to severity as mild nonproliferative diabetic retinopathy (NPDR), moderate NPDR, severe NPDR, and proliferative diabetic retinopathy (PDR). Severe NPDR and PDR are collectively referred to as vision-threatening diabetic retinopathy (VTDR).

## Search strategy

We searched PubMed, Web of Science, and Embase databases up to May 1, 2022, for observational studies that investigated the association between abdominal obesity and DR in patients with diabetes mellitus. The search strategy included the following terms: abdominal obesity, central obesity, visceral obesity, visceral fat, anthropometry, waist circumference, diabetic retinopathy, diabetic eye disease, retinal photographs, optical coherence tomography, and diabetes. The search strategy had no language, publication date, or publication restrictions. Two authors (S.F. and L.Z.) independently screened the initially retrieved articles based on titles and abstracts, at which point any duplicates were removed, and the remaining articles were then sent for full-text review. Any discrepancies regarding inclusion were resolved through group discussions with input from the senior investigator (X.Z.). Details of our search strategy are presented in S1 Table.

## Study selection

Eligible studies had to meet the following criteria: (1) original observational studies; (2) explicitly stated the definition and graded diagnostic criteria for DR; (3) specifically described the measurement or calculation of abdominal obesity parameters, including WC, WHR, WHtR, VFA, LAP, or VAI; (4) evaluated the associations between abdominal obesity and DR in patients with T1DM or T2DM; and (5) reported the mean ± standard deviation (SD) of abdominal obesity parameters or the median and interquartile range (IQR) for conversions available [11]. Studies were excluded if they (1) were animal experiments, case reports, editorials, comments, or literature reviews; (2) repeated reports of the same data in different forms; and (3) contained incomplete data that were still unavailable after contact with the author.

## Data extraction and quality assessment

Data extraction was performed using predefined forms, and the details were as follows: first author, year of publication, country, study design, type of diabetes, severity of diabetic retinopathy, measures of abdominal obesity, number of patients, and mean ± SD of measures. Patients with diabetes without DR were classified into the control group. Data were extracted from each qualified article by two independent investigators (S.F. and L.Z.). Disagreements and uncertainties were resolved by consensus with the third author (X.Z.).

The Agency for Healthcare Research and Quality (AHRQ) recommended criteria were used to evaluate the quality of cross-sectional studies [12]. The criteria consists of 11 items, each of which has an answer of either "yes," "no," or "unclear". The quality of case-control and cohort studies was assessed using the Newcastle-Ottawa Scale (NOS), which contains eight items in three major sections: population selection, comparability, and exposure [13].

## Statistical analysis

Continuous variables were presented as mean ± SD, and standardized mean differences (SMD) with 95% confidence intervals (CI) were calculated. The $I^2$ statistic was used to test for heterogeneity across the studies. A fixed-effects model was used for $I^2 < 50\%$, whereas random-effects models were used for $I^2 \geq 50\%$ [14]. Forest plots were generated for all meta-analyzed outcomes, which were then assessed for statistical significance. To explore potential sources of heterogeneity among the included studies, a considerable number of prespecified subgroup analyses were conducted based on ethnicity, DR severity, and parameters of abdominal obesity. Additionally, sensitivity analyses were performed to identify potential sources of heterogeneity. Publication bias was evaluated by visual inspection of funnel plot asymmetry supplemented by the Egger regression test. Simultaneously, the number of theoretically missing studies was estimated using the trim-and-fill method. We used Stata 16.0 (Stata Corporation, College Station, TX, USA) to perform data analyses. Statistical significance was set at $p$ value $< 0.05$.

## Results

### Literature search

A total of 422 articles were identified after searching databases and checking for duplicates. After reviewing the titles and abstracts, 387 ineligible studies were excluded due to the research topic, article type, and study design. The full texts of 35 potentially related studies were then strictly assessed. Ultimately, 24 studies were included in this meta-analysis. The details of the search methodology and selection process are shown in **Fig 1**.

### Study characteristics and quality evaluation

The 24 studies enrolled 5596 DR patients and 17907 non-DR patients to explore the association between DR and abdominal obesity in patients with diabetes. Seventeen studies were conducted in Asia, four in Europe [15–18] and one each in Australia [19], North America [20] and Africa [21]. All studies were conducted in populations over 18 years. Baseline characteristics of these enrolled studies are shown in **Table 1**. Three case-control studies were of high quality (score $\geq$ 7), and one was fair. All cross-sectional studies had relatively high quality (score $\geq$ 6). The results of quality assessment are presented in **S2** and **S3 Tables**.

### Association between all abdominal obesity evaluation indicators and DR

We applied the random-effects model because of the significant heterogeneity between the included studies ($I^2 = 88.13\%$). We tried to exclude any one study and found that heterogeneity did not change significantly. The results showed a significant difference in abdominal obesity evaluation indicators between the DR and non-DR groups in the T2DM population (SMD 0.12, 95% CI 0.04–0.20, $p < 0.01$, $I^2$ 87.58%). Such statistical differences were also observed in the T1DM population (SMD 0.48, 95% CI 0.11–0.85, $p = 0.04$, $I^2$ 77.03%) (**Fig 2**). Publication bias did not significantly affect the results of this analysis. Further analysis conducted using funnel plot asymmetry estimated that there was some missing literature with small samples and negative results (**S1 Fig**). The sensitivity analyses omitting one study at a time and calculating the pooled risk estimates for the remaining studies, which range from 0.03 (95% CI 0.00, 0.06) to 0.06 (95% CI 0.04, 0.09) showed that no single study had a substantial effect on risk estimates (**Fig 3**).

Furthermore, the respective associations of each abdominal obesity indicator were subjected to a subgroup analysis. The results of the WC subgroup indicated that patients with DR

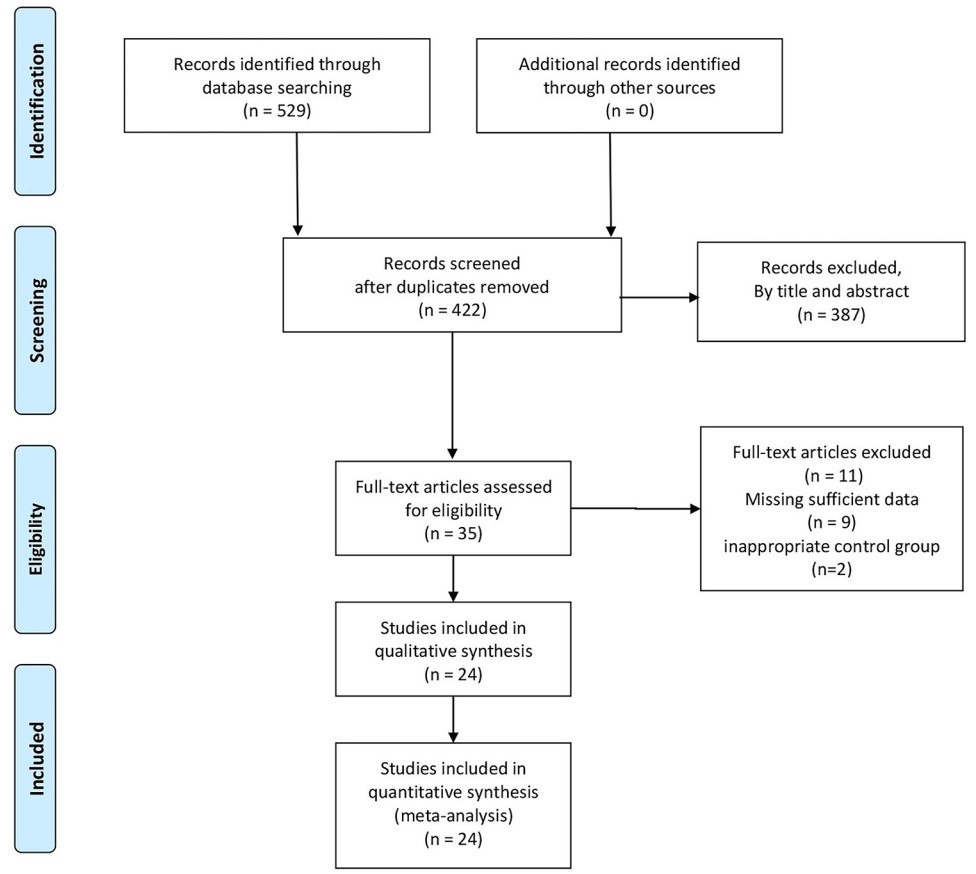

**Fig 1. The PRISMA flow chart.**

had higher WC levels than those without DR (SMD 0.16, 95% CI 0.01–0.31, $p = 0.05$, $I^2$ 89.10%) (**Fig 4A**). The Egger regression test showed no publication bias in the meta-analysis ($p = 0.112$). Meanwhile, the sensitivity analysis presented a robust result that was not influenced by the individual studies (**S2 Fig**).

In the WHR/WHtR subgroup, meta-analysis also indicated the significant association between WHR/WHtR and DR (SMD 0.13, 95% CI 0.02–0.24; $p = 0.02$, $I^2$ 84.32%) (**Fig 4B**). The Egger regression test results showed no significant publication bias ($p = 0.372$). Sensitivity analysis showed that the only study using WHtR as an abdominal obesity indicator by Li et al. [4] had a greater effect on the pooled risk estimates (**S3A Fig**). After this study was removed, sensitivity analysis of the WHR subgroup demonstrated that the observed risk estimate was robust (**S3B Fig**).

Four studies [7,17,28,33] were included in the VFA subgroup. The association of VFA with DR was not statistically significant and was accompanied by high heterogeneity (SMD 0.42, 95% CI -0.17 to 0.97, $p < 0.01$, $I^2$ 93.66%) (**Fig 4C**). Egger's test revealed no publication bias ($p = 0.612$), and sensitivity analysis also provided a robust result (**S4 Fig**).

In addition, two studies [5,9] included in this meta-analysis used VAI as the abdominal obesity evaluation parameter, and three studies [4,9,25] used LAP. The association between VAI and DR was not statistically significant (SMD -0.13, 95% CI -0.22 to -0.03; $p = 0.33$, $I^2$ 0.00%) (**Fig 4D**). In the LAP subgroup, the meta-analysis did not demonstrate any association between LAP and DR (SMD 0.06, 95% CI -0.36 to 0.47; $p < 0.01$, $I^2$ 90.75%).

**Table 1. Characteristics of included studies.**

| Study | country | Study design | Type of diabetes | Severity of diabetic retinopathy | Measures of abdominal obesity | Patients with DR | | | Patients without DR | | |
|---|---|---|---|---|---|---|---|---|---|---|---|
| | | | | | | n | mean | SD | n | mean | SD |
| Wu, 2022 [5] | China | cohort | T2DM | DR | WC | 90 | 91.7 | 8.33 | 8850 | 87.27 | 10.18 |
| | | | | | VAI | 90 | 1.84 | 1.39 | 8850 | 1.89 | 1.57 |
| Chen, 2022 [22] | China | case–control | T2DM | DR | WHR | 1544 | 0.90 | 0.07 | 1544 | 0.89 | 0.07 |
| Yi, 2021 [23] | China | cross-sectional | T2DM | DR | WC | 490 | 89.51 | 9.40 | 1301 | 90.49 | 9.12 |
| | | | | | WHR | 490 | 0.92 | 0.07 | 1301 | 0.92 | 0.06 |
| | | | | | WHtR | 490 | 0.56 | 0.003 | 1301 | 0.57 | 0.06 |
| Li, 2021 [4] | China | cross-sectional | T2DM | m-NPDR | WC | 55 | 91.99 | 6.79 | 61 | 87.87 | 7.77 |
| | | | | VTDR | WC | 53 | 93.34 | 7.63 | 61 | 87.87 | 7.77 |
| | | | | m-NPDR | LAP | 55 | 62.79 | 36.52 | 61 | 49.23 | 34.29 |
| | | | | VTDR | LAP | 53 | 96.35 | 66.03 | 61 | 49.23 | 34.29 |
| Maeda, 2021 [20] | Mexican | case–control | T2DM | DR | WHR | 149 | 0.94 | 0.07 | 149 | 0.91 | 0.07 |
| Wan, 2020 [9] | China | cross-sectional | T2DM | DR | WC | 544 | 90.99 | 9.89 | 1224 | 90.09 | 9.84 |
| | | | | | WHR | 544 | 0.91 | 0.07 | 1224 | 0.91 | 0.07 |
| | | | | | LAP | 544 | 52.22 | 37.22 | 1224 | 54.33 | 39.23 |
| | | | | | VAI | 544 | 2.51 | 1.73 | 1224 | 2.81 | 2.16 |
| Hwang, 2019 [24] | Korea | cross-sectional | T2DM | DR | WC | 185 | 85.27 | 8.01 | 702 | 87.49 | 8.51 |
| Wu, 2019 [25] | China | cross-sectional | T2DM | DR | WC | 76 | 82.61 | 6.55 | 351 | 86.69 | 6.81 |
| | | | | | LAP | 76 | 40.76 | 39.60 | 351 | 55.60 | 44.51 |
| Zhou, 2019 [6] | China | case–control | T2DM | DR | WC | 156 | 92.5 | 7.93 | 156 | 89.9 | 7.40 |
| | | | | | WHR | 156 | 0.94 | 0.06 | 156 | 0.93 | 0.06 |
| Yao, 2019 [26] | China | cross-sectional | T2DM | DR | WHR | 51 | 0.83 | 0.06 | 372 | 0.91 | 0.51 |
| Sasongko, 2018 [27] | Indonesian | cross-sectional | T2DM | m-NPDR | WC | 114 | 89.3 | 10.2 | 671 | 90.9 | 10.3 |
| | | | | VTDR | WC | 258 | 88.8 | 11.7 | 671 | 90.9 | 10.3 |
| Moh, 2018 [28] | Singapore | cross-sectional | T2DM | DR | WC | 241 | 92.6 | 12.1 | 377 | 90.4 | 12.1 |
| | | | | | VFA | 241 | 136.3 | 37.1 | 377 | 126.9 | 39.5 |
| Man, 2016 [29] | Singapore | cross-sectional | T2DM | DR | WHR | 183 | 0.95 | 0.04 | 237 | 0.93 | 0.05 |
| Hu, 2015 [30] | China | cross-sectional | T2DM | DR | WHR | 39 | 0.899 | 0.01 | 290 | 0.894 | 0.03 |
| Rajalakshmi, 2014 [31] | India | cross-sectional | T1DM | DR | WC | 80 | 80.9 | 10.4 | 70 | 73.5 | 10.6 |
| | | | T2DM | DR | WC | 79 | 92.6 | 10.3 | 71 | 89.2 | 11.6 |
| Dossarps, 2014 [17] | France | cross-sectional | T2DM | DR | VFA | 69 | 280.56 | 126.66 | 110 | 286.73 | 117.18 |
| Longo, 2014 [21] | South Africa | case–control | T2DM | DR | WC | 66 | 93.8 | 16.4 | 84 | 95.4 | 12.2 |
| Tomić, 2013 [18] | Croatia | cross-sectional | T2DM | m-NPDR | WC | 19 | 108.21 | 12.09 | 65 | 107.52 | 14.96 |
| | | | | VTDR | WC | 23 | 107.91 | 12.28 | 65 | 107.52 | 14.96 |

*(Continued)*

**Table 1.** (Continued)

| Study | country | Study design | Type of diabetes | Severity of diabetic retinopathy | Measures of abdominal obesity | Patients with DR | | | Patients without DR | | |
|---|---|---|---|---|---|---|---|---|---|---|---|
| | | | | | | n | mean | SD | n | mean | SD |
| | | | | m-NPDR | WHR | 19 | 0.96 | 0.07 | 65 | 0.96 | 0.08 |
| | | | | VTDR | WHR | 23 | 0.97 | 0.07 | 65 | 0.96 | 0.08 |
| Dirani, 2011 [19] | Australia | cross-sectional | T2DM | DR | WC | 321 | 107.7 | 15.3 | 171 | 104.7 | 17.5 |
| | | | | | WHR | 321 | 0.98 | 0.08 | 171 | 0.96 | 0.09 |
| Anan, 2010 [7] | Japan | cross-sectional | T2DM | DR | WC | 31 | 91.7 | 9.8 | 71 | 84.3 | 8.5 |
| | | | | | VFA | 31 | 162 | 62 | 71 | 87 | 30 |
| Zhang, 2009 [32] | China | cross-sectional | T2DM | DR | WHR | 78 | 0.88 | 0.06 | 313 | 0.89 | 0.06 |
| van Leiden, 2003 [15] | Netherlands | cross-sectional | T2DM | DR | WC | 27 | 97.8 | 10.1 | 206 | 92.1 | 10.8 |
| | | | | | WHR | 27 | 0.96 | 0.07 | 206 | 0.91 | 0.09 |
| Asakawa, 2002 [33] | Japan | cross-sectional | T2DM | m-NPDR | VFA | 20 | 58.1 | 37.5 | 126 | 83.3 | 69.8 |
| | | | | VTDR | VFA | 29 | 83.8 | 47.3 | 126 | 83.3 | 69.8 |
| Chaturvedi, 2001 [16] | UK | cross-sectional | T1DM | DR | WHR | 429 | 0.87 | 0.12 | 335 | 0.83 | 0.13 |

Abbreviations: T1DM, type 1 diabetes mellitus; T2DM, type 2 diabetes mellitus; DR, diabetic retinopathy of any degree; m-NPDR, mild or moderate nonproliferative DR; VTDR, vision-threatening DR; WC, waist circumference; WHR, waist-hip ratio; WHtR, waist-height ratio; VFA, visceral fat area; LAP, lipid accumulation product; VAI, visceral adiposity index; SD, standard deviation.

## Association between abdominal obesity and DR of varying severity

Four of the included studies [4,18,27,33] separately examined the association between abdominal obesity and diabetic retinopathy of varying severity, including mild or moderate NPDR and VTDR. The results of the meta-regression analysis ($p = 0.558$) suggested that DR severity did not lead to high heterogeneity in this meta-analysis (**S5A Fig**). There was no statistically significant difference in the association between abdominal obesity and VTDR compared to that with m-NPDR (test of group differences: $p = 0.49$) (**Fig 5**).

## Association between abdominal obesity and DR in different ethnicity

We performed a subgroup analysis based on ethnicity (Asian and Caucasian) and found a statistically significant difference between the two groups (test of group differences: $p = 0.04$). The association between abdominal obesity and DR in Caucasians (SMD 0.26, 95% CI 0.15–0.37, $p = 0.11$) was stronger than that in Asians (SMD 0.11, 95% CI 0.02–0.21, $p < 0.01$) (**Fig 6**). However, this distinction became statistically insignificant when tested with meta-regression ($p = 0.128$) (**S5B Fig**). Because the heterogeneity of included studies in Caucasian populations ($I^2$ 39.11%) is much lower than that of studies in Asian populations ($I^2$ 88.40%), ethnicity was considered a significant contributor to heterogeneity in this meta-analysis.

## Discussion

### Main findings

This meta-analysis included 5596 patients with DR and 17907 diabetic patients without DR from 24 studies. The salient finding of our study was that abdominal obesity, measured by WC

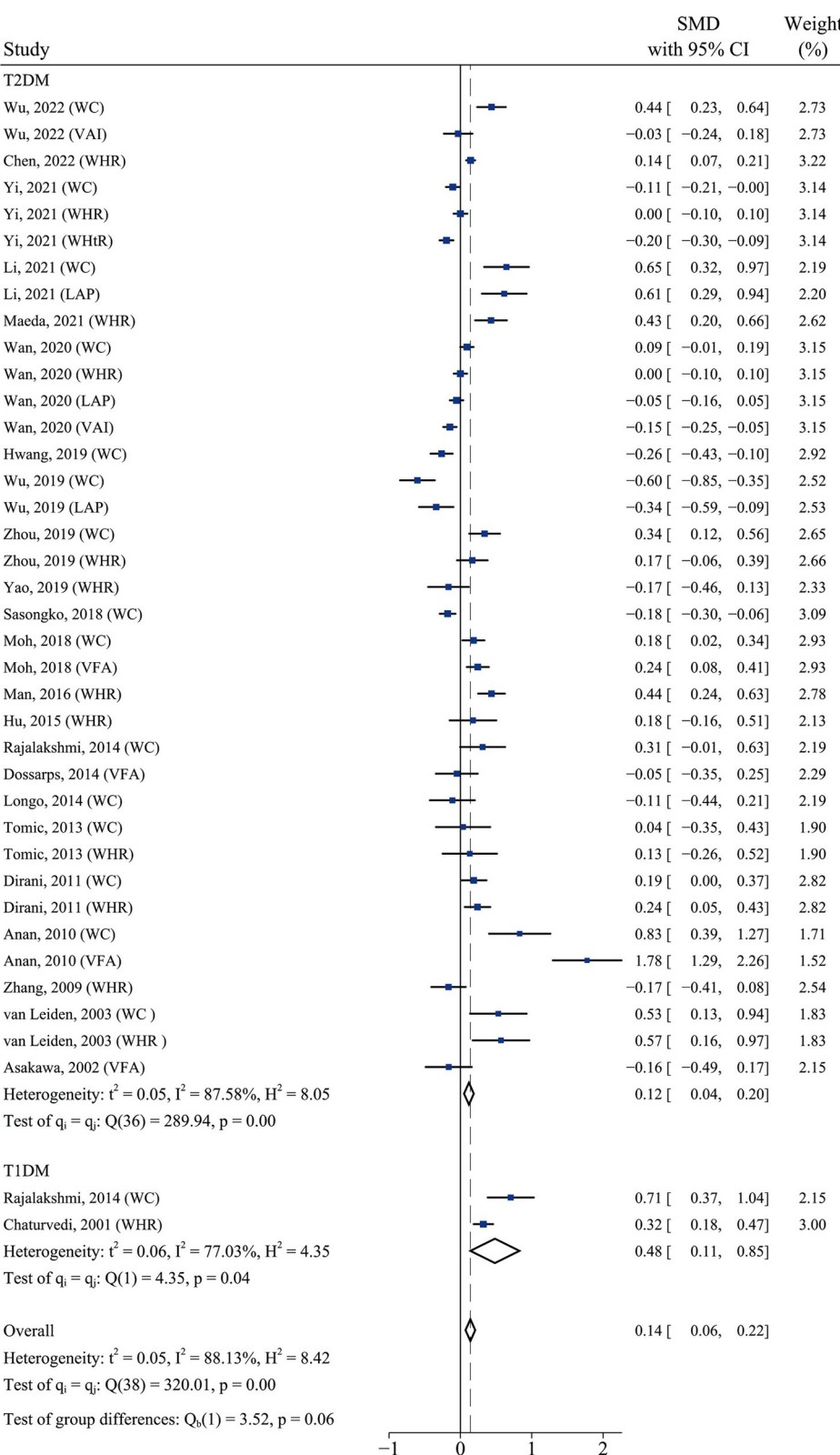

**Fig 2. Forest plot of the association between abdominal obesity and DR in T1DM, and T2DM patients.**
Abbreviations: T1DM, type 1 diabetes mellitus; T2DM, type 2 diabetes mellitus; WC, waist circumference; WHR, waist-hip ratio; WHtR, waist-height ratio; VFA, visceral fat area; LAP, lipid accumulation product; VAI, visceral adiposity index; SMD, standardised mean differences; CI, confidence intervals; P, probability value.

and WHR, was associated with DR in patients with type 1 and type 2 diabetes. This association was stronger among Caucasians compared to Asians.

## Association between abdominal obesity and DR

The American Academy of Ophthalmology guidelines [34] conservatively mention a trend for stepwise increases in DR, corresponding to the number of MS components. Our study further clarified that abdominal obesity is independently associated with DR. The implications of these findings are quite important, since they may help illustrate that reducing abdominal obesity has a positive impact on the prevention of DR in patients with DM, regardless of ethnicity.

Despite the current clinical literature examining the association between abdominal obesity and DR, conflicting results have been reported. Our results for the WC subgroup concur with those of the most recent meta-analyses [6], which demonstrated that abdominal obesity defined by WC is associated with the risk of DR. However, our results update their findings regarding WHR, for which the former study did not find a statistically significant association. Therefore, screening programs for WC and WHR may be more helpful in identifying patients

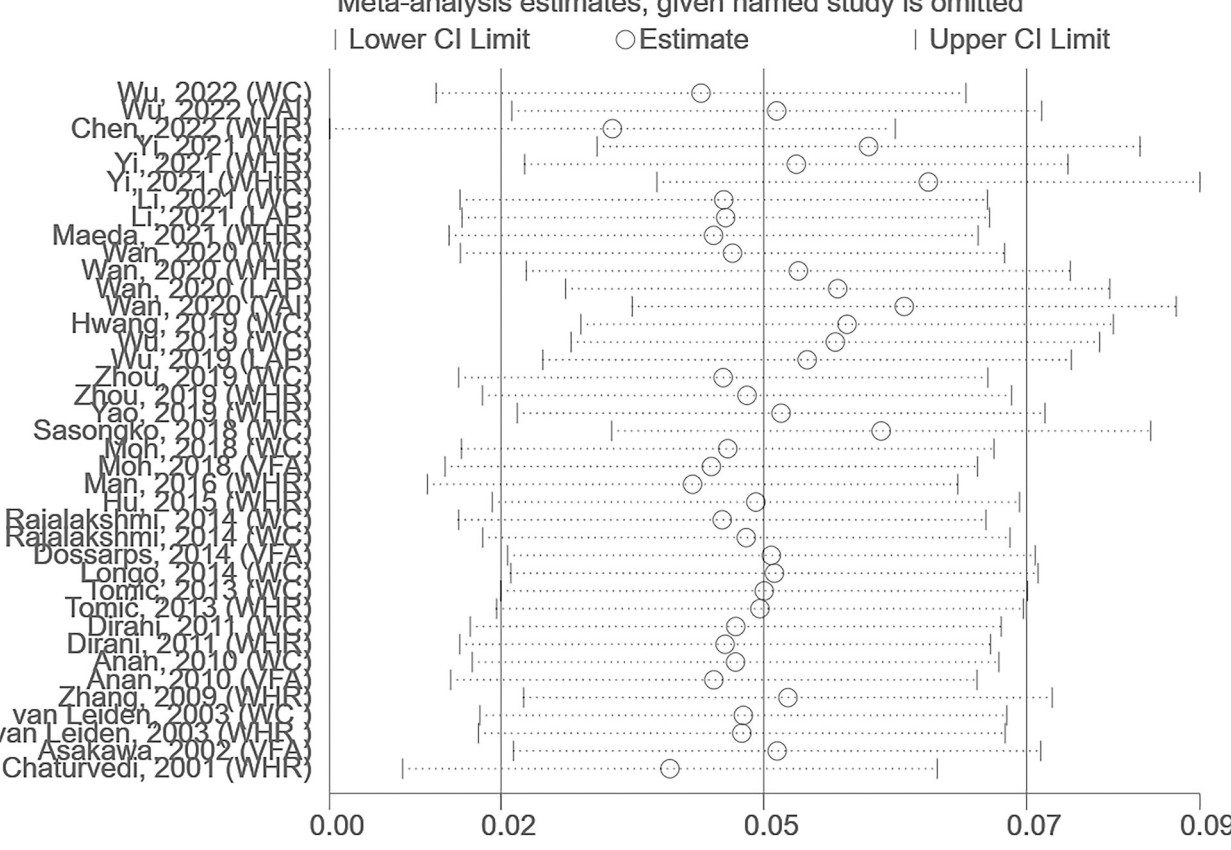

**Fig 3. Sensitivity analysis of included 24 studies.** Abbreviations: CI, confidence intervals.

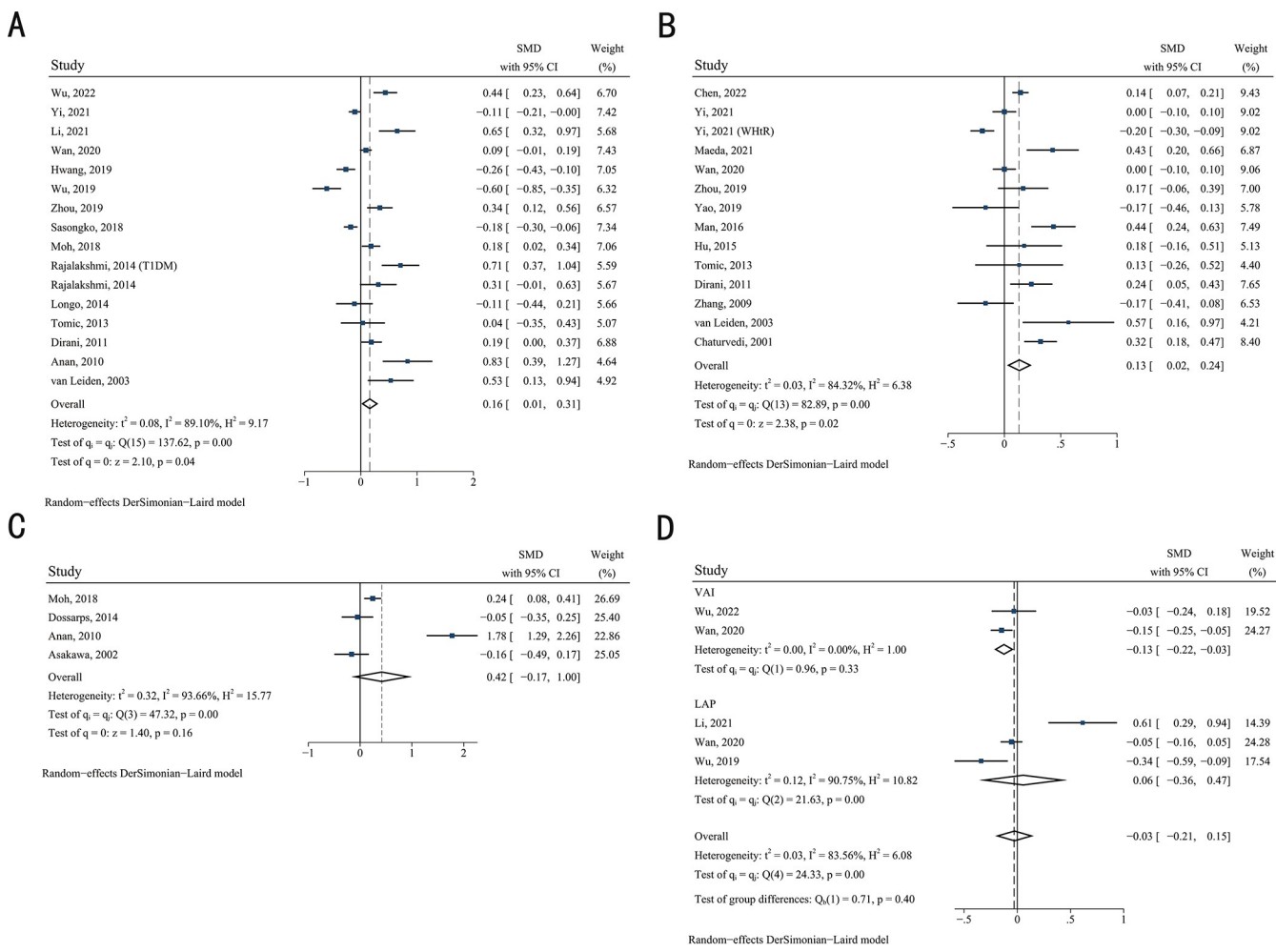

**Fig 4. Forest plot of abdominal obesity evaluation indicator subgroups.** (A) Forest plot of the association between waist circumference and DR. (B) Forest plot of the association between waist-hip ratio and DR. (C) Forest plot of the association between visceral fat area and DR. (D) Forest plot of the association between VAI, LAP and DR. Abbreviations: SMD, standardised mean differences; CI, confidence intervals; P, probability value.

at higher risk for DR progression than VFA, VAI, and LAP. Furthermore, considering economic expenditure, direct measurements of WC and WHR are more suitable for widespread screening.

The meta-analysis showed significant heterogeneity in the combined group and in each subgroup of abdominal obesity evaluation indicators. Therefore, it is important to explore and assess the heterogeneity of the results. One possible cause of this variation in study effect size may be ethnic differences in study inclusion. For instance, in the subgroup analysis that included only Caucasians, heterogeneity was reduced by 49.02% compared to the overall analysis. This indicates that the association between abdominal obesity and DR in Caucasians is more robust than that in Asians. Interestingly, this ethnically distinct feature was also present in studies on the association between DR and generalized obesity defined by BMI in a different format. Numerous clinical evidence suggested that BMI and DR tended to be insignificantly or negatively associated in Asian DM populations [35,36]. In contrast, several studies among Caucasians have reported a significant positive association between high BMI and DR [37,38]. Caucasian patients with DM generally have a high BMI [39], and the effect of abdominal obesity on DR is highlighted when abdominal fat accumulation occurs. In contrast, Asians with

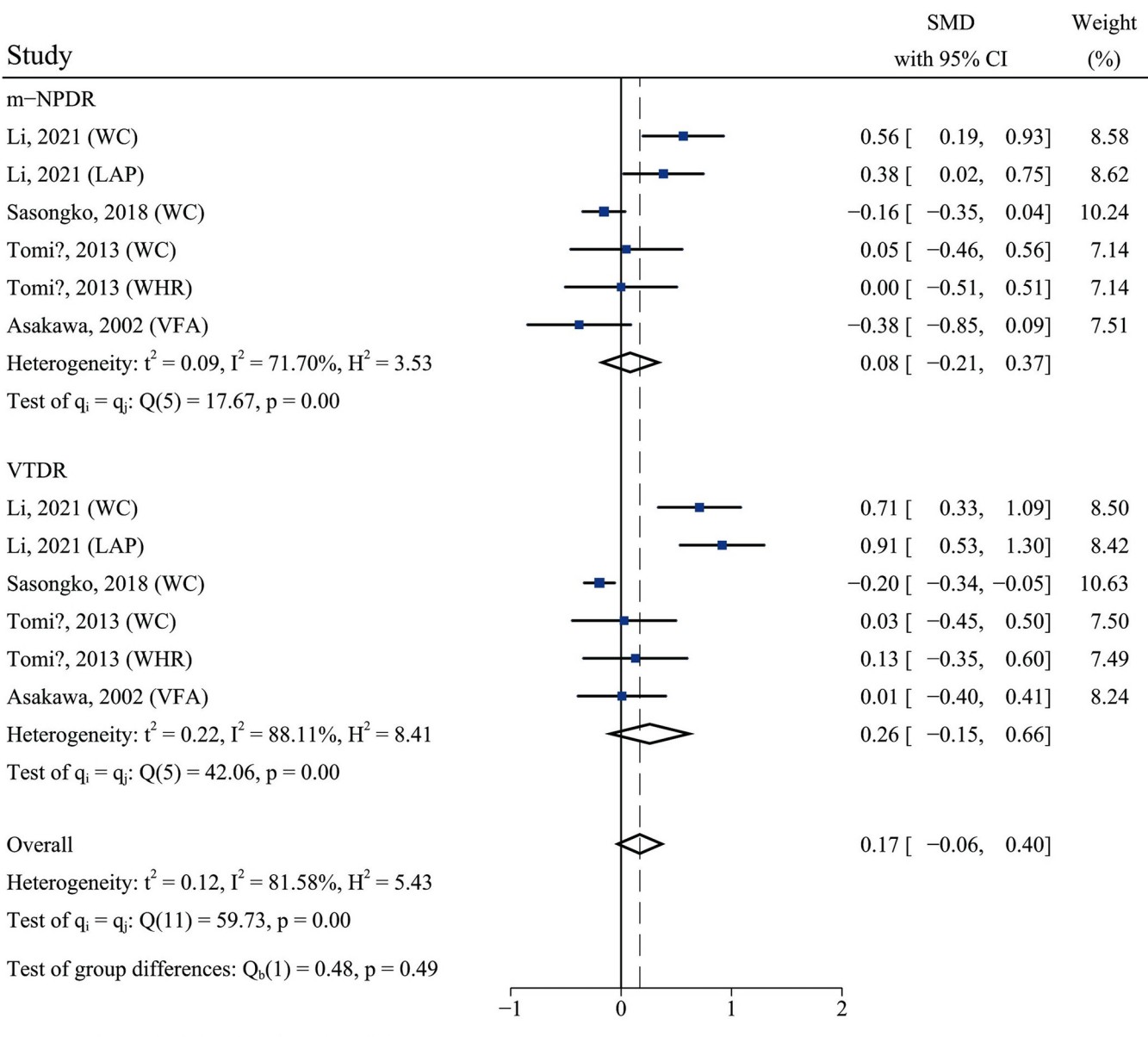

**Fig 5. Forest plot of the association between abdominal obesity and diabetic retinopathy severity.**

lower BMI ($<25$ kg/m$^2$) [40] also suffered from visceral fat accumulation and the potential risk of DR. So, the confusion about the "types of obesity" probably leads to the erroneous conclusion that BMI and DR are not related. Only two BMI-stratified studies [22,41] that reported association between abdominal obesity and DR in Asian populations suggested that high WC with normal BMI and high WHR accompanied with low to moderate BMI levels are significantly associated with DR, supporting our view. Whether these differences in ethnicity are exactly related to differences in abdominal fat accumulation is a question that remains to be answered by future research.

Abdominal obesity, especially in patients with BMI $<$ 25 kg/m$^2$, is characterized by excessive accumulation of visceral fat in the abdominal cavity as a core pathology. Computed tomography in the umbilical plane is generally considered the "gold standard" for assessing

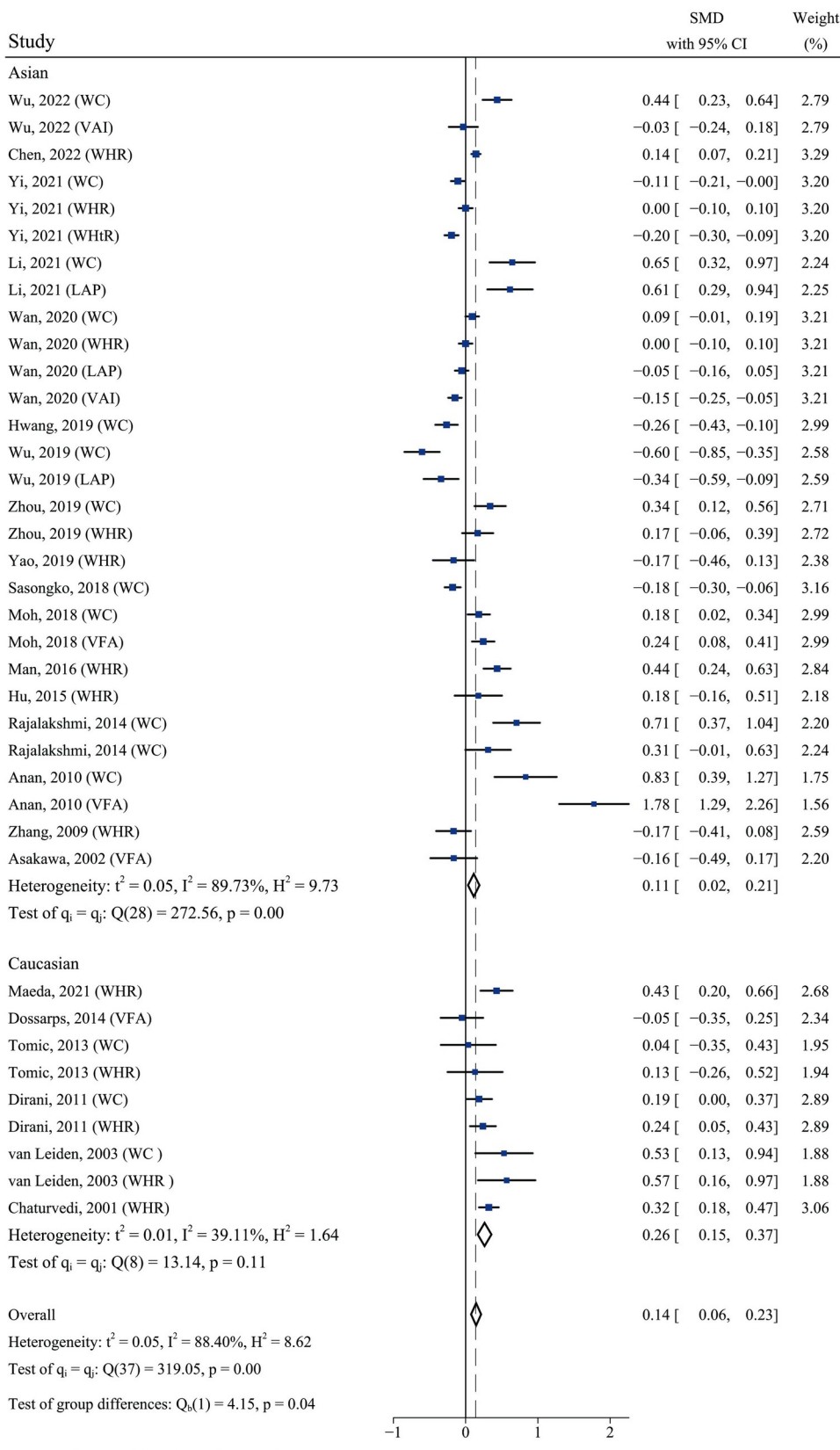

**Fig 6. Forest plot of the association between abdominal obesity and DR in different ethnic subgroups.**

VFA [42]. In the VFA subgroup, Anan et al. [7] measured VFA using CT and found that VFA was an independent predictor of DR risk in Japanese T2DM patients. Moh, A et al. [28] similarly found that VFA, assessed by bioelectrical impedance analysis, was a more robust risk factor for DR than WC in a multi-ethnic Asian population. VFA, compared to WC, was not affected by the confounding effect of subcutaneous fat. This further suggests that the association between abdominal obesity and DR is related to visceral fat accumulation in humans. Further prospective studies with large sample sizes and rigorous controls are required to definitively validate the conclusions, as the combined results of the VFA subgroup showed a non-significant association. This is probably related to the high heterogeneity of subgroup analysis because three different measurement methods (CT [33], MRI [17] and bioelectrical impedance analysis) were used in the four studies with smaller sample sizes.

The underlying pathological mechanism of the association between abdominal obesity and DR has not been fully elucidated. Previous studies have suggested that it may be implicated in insulin resistance and adipose tissue inflammation due to visceral fat accumulation. M1 macrophage polarization and recruitment within the pathologically accumulated adipose tissue promotes the expression of inflammatory factors, such as tumor necrosis factor-$\alpha$ and interleukin 6 [43,44], further damaging the capillary endothelium. In a meta-analysis by Zhao et al. [45], abdominal obesity was significantly associated with diabetic kidney disease, another microangiopathy of DM. Studies have found that chronic inflammation can also lead to retinal capillary occlusion, inducing retinal ischemia and retinopathy [46]. Meanwhile, abdominal obesity increases insulin resistance in patients with diabetes [47], which further causes retinal microvascular ischemia, hypoxia, and oxidative stress damage to the vessel wall, leading to DR. In addition, studies by Shimajiri et al. [48] and Gao et al. [49] found that metabolic syndrome was also significantly positively associated with the development of DR, while the risk increased with the number of MS components. These hypotheses on the mechanism of abdominal obesity require more in-depth clinical and basic verification.

## Strengths and limitations

The strengths of this study are as follows: First, our work is the first to systematically evaluate and meta-analyze the association between various parameters of abdominal obesity and DR in diabetic patients worldwide. Second, compared with previous meta-analyses among Chinese populations evaluated by WC and WHR, we included a broader population and more evaluation indicators. Finally, we assessed the effects of different parameters, DR severity, and ethnicity on outcomes using subgroup analysis and explored potential sources of heterogeneity.

However, this study has some limitations. The diagnostic criteria for abdominal obesity are not the same for men and women, and the association between obesity and DR may vary according to sex. However, the study failed to adequately assess gender as a potentially important source of heterogeneity because of incomplete data from the original studies. In addition, DR is influenced by many factors, such as the duration of DM and glucose and lipid levels. Nevertheless, each study adjusted for different confounding factors; therefore, the conclusions of this meta-analysis could not wholly exclude confounding factors. Moreover, the literature included in the study was mainly cross-sectional, which could not provide sufficient evidence to determine the causal relationship between abdominal obesity and DR. Further large sample sizes and prospective cohort studies are yet to be conducted to clarify the conclusions.

## Conclusion

This meta-analysis concluded that abdominal obesity, measured by WC and WHR, was associated with DR in patients with type 1 and type 2 diabetes. This association is stronger in

Caucasians than in Asians. In addition, isolated abdominal obesity may be more associated with DR. Further large-sampled, prospective cohort studies are yet to be conducted to clarify these findings. No differences were detected in the association between abdominal obesity and the different degrees of diabetic retinopathy. No significant association was found between CAI, LAP, and DR.

## Supporting information

**S1 Checklist.**
(DOCX)

**S1 Fig. The funnel plot.** Abbreviations: SMD, standardised mean differences; P, probability value.
(TIF)

**S2 Fig. Sensitivity analysis of WC subgroup.** Abbreviations: WC, waist circumference; CI, confidence intervals.
(TIF)

**S3 Fig. Sensitivity analysis of WHR/WHtR subgroup.** (A) Sensitivity analysis including WHR and WHtR. (B) Sensitivity analysis of subgroup after excluding WHtR. Abbreviations: WHR, waist-hip ratio; WHtR, waist-height ratio; CI, confidence intervals.
(TIF)

**S4 Fig. Sensitivity analysis of VFA subgroup.** Abbreviations: VFA, visceral fat area; CI, confidence intervals.
(TIF)

**S5 Fig. Meta-regression analysis.** (A) meta-regression analysis of DR severity. (B) meta-regression analysis of ethnicity.
(TIF)

**S1 Table. Search strategy of Pubmed database.**
(DOCX)

**S2 Table. Quality assessment of cross-sectional studies according to AHRQ recommended criteria.**
(DOCX)

**S3 Table. Quality assessment of case-control and cohort studies according to NOS.**
(DOCX)

## Author Contributions

**Data curation:** Liwei Zhang.

**Formal analysis:** Shouqiang Fu, Jing Xu.

**Investigation:** Liwei Zhang, Jing Xu.

**Methodology:** Xiaoyun Zhu.

**Project administration:** Ximing Liu.

**Software:** Liwei Zhang.

**Supervision:** Ximing Liu.

**Writing – original draft:** Shouqiang Fu.

**Writing – review & editing:** Ximing Liu, Xiaoyun Zhu.

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
