## [Decision Letter · Decision Letter 0]

24 Oct 2022

PONE-D-22-22136Association between Abdominal Obesity and Diabetic Retinopathy in Patients with Diabetes Mellitus: Systematic Review and Meta-analysisPLOS ONE

Dear Dr. Liu,

Thank you for submitting your manuscript to PLOS ONE. After careful consideration, we feel that it has merit but does not fully meet PLOS ONE’s publication criteria as it currently stands. Therefore, we invite you to submit a revised version of the manuscript that addresses the points raised during the review process.

We look forward to receiving your revised manuscript.

Kind regards,

Tariq Jamal Siddiqi

Academic Editor

PLOS ONE

Journal Requirements:

Reviewers' comments:

Reviewer's Responses to Questions

**Comments to the Author**

1. Is the manuscript technically sound, and do the data support the conclusions?

Reviewer #1: Yes

2. Has the statistical analysis been performed appropriately and rigorously? 

Reviewer #1: Yes

3. Have the authors made all data underlying the findings in their manuscript fully available?

Reviewer #1: Yes

4. Is the manuscript presented in an intelligible fashion and written in standard English?

Reviewer #1: Yes

5. Review Comments to the Author

Reviewer #1: Liu et al. conducted a study on “Association between Abdominal Obesity and Diabetic Retinopathy in Patients with Diabetes Mellitus: Systematic Review and Meta-analysis”, in which they explored the association between different parameters of abdominal obesity and the risk of diabetic retinopathy in diabetes mellitus patients. They found that waist circumference and waist-hip ratio were significant predictors of diabetic retinopathy while visceral fat area, lipid accumulation product and visceral adiposity index were not.

In my opinion, this study may be improved by incorporating the following edits:

1. The authors frequently use acronyms such as DR, WC, WHR, WHtR, etc without first defining them in the text. The first usage of an acronym within the full-text should always be defined with its full form alongside the acronym in brackets. Please make these changes for all acronyms used in the article. Definitions made in this form in the abstract do not carry forward into the main text of the article and must be redefined.

2. The authors should consider rephrasing lines 42-44 to “Currently recognized risk factors affecting the development of DR include the duration of diabetes, elevated HbA1c, blood glucose, blood pressure, serum cholesterol, and low-density lipoprotein levels.”

3. In lines 58-59, the authors should consider removing the sentence, “Therefore, studies further clarifying the association between abdominal obesity and DR in diabetic patients is essential.” This sentence is not well-suited for the introduction and would be more appropriately placed in the discussion section if the authors feel that some aspects of the association they are exploring within this article are not adequately clarified despite the them performing a meta-analysis to this end.

4. The authors should clearly state within the introduction section that they are performing this meta-analysis to synthesize all current data and reach a definitive conclusion in order to address the contradictory and ambiguous current literature and clarify the situation at hand.

5. The authors should also clearly state the aims of this meta-analysis at the end of the introduction section. They should state the outcomes they seek to explore and ambiguities in the data they hope to resolve.

6. While the authors have provided adequate background knowledge regarding the field at hand in the introduction section, they have not appropriately contextualized the study in light of previous systematic reviews and meta-analyses exploring this association. Do previous systematic reviews or meta-analyses exploring this topic exist? If so, the authors should mention them and highlight why they believed it necessary to update them, and what additional utility the current study hopes to provide. If not, the authors should clearly state that this is the first study of its kind.

7. The authors should take care to highlight the novel features of the analyses they are undertaking in the introduction section. What makes this study unique and different from other studies exploring this association? Are the authors exploring any unique outcomes or do they perhaps hope to utilize any new analyses which would improve the accuracy and reliability of previous results?

8. In the methodology section, please clearly state what comparison or control group the authors are referring to for this study.

9. In lines 89-90, the authors state that “The search process was performed independently by two authors (SF and LZ).” Please clearly state how these authors resolved any disagreements.

10. In the search strategy portion of the methodology section, the authors should clarify if the search was performed with any language restrictions. If so, the authors should provide a compelling reason for this restriction. If not, the authors should clearly state that the search was performed with no language restrictions.

11. Please change lines 115-118 to “The quality of case-control and cohort studies was assessed by the Newcastle-Ottawa Scale (NOS), which contains eight items in three major sections: population selection, comparability, and exposure[18].”

12. In the search strategy portion of the methodology section, please clearly state that the articles were first screened on the basis of title and abstract, at which point any duplicates were removed, and the remaining articles were then sent for a full text review.

13. In line 122-123, the authors state that they used the I2 statistic to distinguish between using a random- and a fixed- effects model for their analysis. A suitable reference should be provided, justifying the usage of the I2 statistic in this manner.

14. In the methodology section, please clearly state that the analysis was used to generate forest plots which were then assessed for significance.

15. The “measurements and definitions” portion of the methodology section is somewhat ambiguous and confusing. Did they authors utilize these formulae in the present analysis? If so, then these formulae would be better placed in the statistical analysis section. If not, then extraneous information like the formulae should be removed from this section as they provide no additional utility and are not pertinent to the study at hand.

16. In line 141 please consider highlighting Figure 1 as the PRISMA flowchart.

17. In the results section, please add a summary statement about the results of the quality assessment. How many articles included in the analysis were of high methodological quality?

18. Lines 153-155 from “The results indicated….” are very poorly phrased which leads to significant ambiguity and confusion for the reader. The authors should revise and amend these lines for the sake of clarity and comprehension.

19. In lines 157-158, instead of stating that the “P value of Egger’s test was greater than 0.05”, the authors should simply mention that there was no publication bias significantly influencing the results of this analysis.

20. Please rephrase line 163 to “….. showed that no single study had a substantial effect on risk estimates”

21. The authors should also explore whether any significant heterogeneity was attributed to any single study and was perhaps resolved to the point of non-significance upon its removal.

22. Lines 164-166 “A total of six abdominal obesity evaluation indicators included in this meta-analysis were WC, WHR, WHtR, VFA, LAP, and VAI in 24 studies.” should be removed as they provide no additional information beyond what has already been provided previously.

23. Please provide figure and table legends and title and label all figures and legends appropriately within the text.

24. The authors should also provide a legend or index for the supplementary information.

25. Lines 190-191, please refrain from making statements commenting on the positivity or negativity of associations for statistically non-significant outcomes.

26. In the discussion section, please highlight the novelty of your findings. Were the results of the analysis unique and were they different from previous such analyses? If so, please provide a valid reason for this change in results. If not, please highlight the additional utility your findings may provide, either in the form of additional validation or simply being more up to date with current literature.

27. In the discussion section the authors should discuss the significance of their findings. How were these findings important? What do they demonstrate that may alter guidelines or inspire change in clinical or research practices in this field. The authors should also specifically mention any new avenues of research that have opened up as a result of this study.

28. The article contains many grammatical and language errors. While the most significant ones have been highlighted above, many still remain which make the article confusing for the reader. The authors should revise the writing and should take care to amend these grammatical and language errors for the sake of readability and comprehension.

6. PLOS authors have the option to publish the peer review history of their article (what does this mean?). If published, this will include your full peer review and any attached files.

Reviewer #1: **Yes: **Muhammad Talha Maniya

---

## [Author Response · Author response to Decision Letter 0]

16 Nov 2022

Dear Editor:

We thank you and the reviewers for your thoughtful suggestions and insights. The manuscript has benefited from these insightful suggestions. I look forward to working with you and the reviewers to move this manuscript closer to publication in PLoS ONE. The manuscript has been rechecked and the necessary changes have been made in accordance with the reviewers’ suggestions. Responses to the comments and changes in the revised manuscript are as follows: 

Question 1:

The authors frequently use acronyms such as DR, WC, WHR, WHtR, etc without first defining them in the text. The first usage of an acronym within the full-text should always be defined with its full form alongside the acronym in brackets. Please make these changes for all acronyms used in the article. Definitions made in this form in the abstract do not carry forward into the main text of the article and must be redefined.

Answer 1:

We have modified all the acronyms used in the main text of the article.

Question 2:

The authors should consider rephrasing lines 42-44 to “Currently recognized risk factors affecting the development of DR include the duration of diabetes, elevated HbA1c, blood glucose, blood pressure, serum cholesterol, and low-density lipoprotein levels.”

Answer 2:

We have corrected the grammatical errors in this sentence based on the reviewers’ comments.

(Line 55 in Introduction: Currently recognized risk factors affecting the development of DR include the duration of diabetes, elevated HbA1c, blood glucose, blood pressure, serum cholesterol, and low-density lipoprotein levels.)

Question 3:

In lines 58-59, the authors should consider removing the sentence, “Therefore, studies further clarifying the association between abdominal obesity and DR in diabetic patients is essential.” This sentence is not well-suited for the introduction and would be more appropriately placed in the discussion section if the authors feel that some aspects of the association they are exploring within this article are not adequately clarified despite the them performing a meta-analysis to this end.

Answer 3:

We revised the previous inappropriate narrative and re-edited the introduction.

Question 4:

The authors should clearly state within the introduction section that they are performing this meta-analysis to synthesize all current data and reach a definitive conclusion in order to address the contradictory and ambiguous current literature and clarify the situation at hand.

Answer 4:

We have described the method for performing this meta-analysis based on the reviewers' suggestions.

(Line 83 in Introduction: Therefore, given the paucity of evidence and limitations of previous studies, we carried out this updated meta-analysis to further evaluate the association)

Question 5:

The authors should also clearly state the aims of this meta-analysis at the end of the introduction section. They should state the outcomes they seek to explore and ambiguities in the data they hope to resolve.

Answer 5:

We have stated the aims of this meta-analysis at the end of the Introduction.

(Line 84 in Introduction: We carried out this updated meta-analysis to further evaluate the association between abdominal obesity parameters (WC, WHR, LAP, VFA, and VAI) and DR. Furthermore, we assessed the potential effects of different patient ethnicities, DR severity, and types of diabetes on the outcomes.)

Question 6:

While the authors have provided adequate background knowledge regarding the field at hand in the introduction section, they have not appropriately contextualized the study in light of previous systematic reviews and meta-analyses exploring this association. Do previous systematic reviews or meta-analyses exploring this topic exist? If so, the authors should mention them and highlight why they believed it necessary to update them, and what additional utility the current study hopes to provide. If not, the authors should clearly state that this is the first study of its kind.

Answer 6:

In the Introduction section, we have supplemented the previous meta-analyses exploring this association and detailed its limitations to illustrate that our updated meta-analysis is necessary.

(Line 72 in Introduction: A previous meta-analysis evaluating the association between abdominal obesity and DR among the Chinese population revealed that abdominal obesity defined by waist circumference (WC) is associated with the risk of DR, while the waist-hip ratio (WHR) is not. However, they did not compare the results among different ethnicities. Additionally, their analysis did not explore whether DR of different severities has an equal association with abdominal obesity.)

Question 7:

The authors should take care to highlight the novel features of the analyses they are undertaking in the introduction section. What makes this study unique and different from other studies exploring this association? Are the authors exploring any unique outcomes or do they perhaps hope to utilize any new analyses which would improve the accuracy and reliability of previous results?

Answer 7:

We have highlighted the novel features of the analysis in the introduction section based on the reviewers' suggestions.

(Line 78 in Introduction: In addition, several recent clinical studies have used new measurement parameters to define abdominal obesity, such as lipid accumulation product (LAP), Visceral fat area (VFA) and visceral adiposity index (VAI). Currently, no meta-analysis have incorporated these parameters to investigate the association between abdominal obesity and DR.)

Question 8:

In the methodology section, please clearly state what comparison or control group the authors are referring to for this study.

Answer 8:

We have added a definition of the control group to the data extraction section. 

(Line 154 in Methods: Patients with diabetes without DR were classified into the control group.)

Question 9:

In lines 89-90, the authors state that “The search process was performed independently by two authors (SF and LZ).” Please clearly state how these authors resolved any disagreements.

Answer 9:

We have added a solution to the disagreement in the Search Strategy section. 

(Line 132 in Methods: Any discrepancies regarding inclusion were resolved through group discussions with input from the senior investigator (X.Z.).)

Question 10:

In the search strategy portion of the methodology section, the authors should clarify if the search was performed with any language restrictions. If so, the authors should provide a compelling reason for this restriction. If not, the authors should clearly state that the search was performed with no language restrictions.

Answer 10:

We have added this information to the search strategy section.

(Line 127 in Methods: The search strategy had no language, publication date, or publication restrictions.)

Question 11:

Please change lines 115-118 to “The quality of case-control and cohort studies was assessed by the Newcastle-Ottawa Scale (NOS), which contains eight items in three major sections: population selection, comparability, and exposure[18].”

Answer 11:

We have modified this sentence based on the reviewer's comment.

(Line 162 in Methods: The quality of case-control and cohort studies was assessed using the Newcastle-Ottawa Scale (NOS), which contains eight items in three major sections: population selection, comparability, and exposure.)

Question 12:

In the search strategy portion of the methodology section, please clearly state that the articles were first screened on the basis of title and abstract, at which point any duplicates were removed, and the remaining articles were then sent for a full text review.

Answer 12:

We have added this information to the search strategy section.

(Line 128 in Methods: Two authors (S.F. and L.Z.) independently screened the initially retrieved articles based on titles and abstracts, at which point any duplicates were removed, and the remaining articles were then sent for full-text review.)

Question 13:

In line 122-123, the authors state that they used the I2 statistic to distinguish between using a random- and a fixed- effects model for their analysis. A suitable reference should be provided, justifying the usage of the I2 statistic in this manner.

Answer 13:

We have cited relevant literature to justify the statistical approach.

(Line 511 in References: Duan J Y, Zheng W H, Zhou H, et al. Energy delivery guided by indirect calorimetry in critically ill patients: a systematic review and meta-analysis[J]. Crit Care, 2021,25(1):88.)

Question 14:

In the methodology section, please clearly state that the analysis was used to generate forest plots which were then assessed for significance.

Answer 14:

We have added an explicit description of the forest plots in the Statistical Analysis section.

(Line 174 in Methods: Forest plots were generated for all meta-analyzed outcomes, which were then assessed for statistical significance.)

Question 15:

The “measurements and definitions” portion of the methodology section is somewhat ambiguous and confusing. Did they authors utilize these formulae in the present analysis? If so, then these formulae would be better placed in the statistical analysis section. If not, then extraneous information like the formulae should be removed from this section as they provide no additional utility and are not pertinent to the study at hand.

Answer 15:

We have removed the formulae based on the reviewer’s comments.

Question 16:

In line 141 please consider highlighting Figure 1 as the PRISMA flowchart.

Answer 16:

We have revised the description in Figure 1.

(Line 197 in Results: Fig 1 The PRISMA flow chart.)

Question 17:

In the results section, please add a summary statement about the results of the quality assessment. How many articles included in the analysis were of high methodological quality?

Answer 17:

We have added a summary statement regarding the results of the quality assessment in the Results section.

(Line 207 in Results: Three case-control studies were of high quality (score ≥ 7), and one was fair. All cross-sectional studies had relatively high quality (score ≥ 6).)

Question 18:

Lines 153-155 from “The results indicated….” are very poorly phrased which leads to significant ambiguity and confusion for the reader. The authors should revise and amend these lines for the sake of clarity and comprehension.

Answer 18:

We fixed this ambiguity caused by terrible grammatical errors.

(Line 215 in Results: The results showed a significant difference in abdominal obesity evaluation indicators between the DR and non-DR groups in the T2DM population (SMD 0.12, 95% CI 0.04-0.20, p < 0.01, I2 87.58%). Such statistical differences were also observed in the T1DM population (SMD 0.48, 95% CI 0.11-0.85, p = 0.04, I2 77.03%).)

Question 19:

In lines 157-158, instead of stating that the “P value of Egger’s test was greater than 0.05”, the authors should simply mention that there was no publication bias significantly influencing the results of this analysis.

Answer 19:

We have re-described the statistical results according to the reviewers' suggestions.

(Line 224 in Results: Publication bias did not significantly affect the results of this analysis.)

Question 20:

Please rephrase line 163 to “….. showed that no single study had a substantial effect on risk estimates”

Answer 20:

We have modified this grammatical error based on the reviewer's comment.

(Line 230 in Results: ……showed that no single study had a substantial effect on risk estimates.)

Question 21:

The authors should also explore whether any significant heterogeneity was attributed to any single study and was perhaps resolved to the point of non-significance upon its removal.

Answer 21:

We tried to exclude any studies and found no significant changes in heterogeneity. Nevertheless, we have added this result to the Results section based on the reviewer's comment.

(Line 213 in Results: We tried to exclude any one study and found that heterogeneity did not change significantly.)

Question 22:

Lines 164-166 “A total of six abdominal obesity evaluation indicators included in this meta-analysis were WC, WHR, WHtR, VFA, LAP, and VAI in 24 studies.” should be removed as they provide no additional information beyond what has already been provided previously.

Answer 22:

We deleted this description.

Question 23:

Please provide figure and table legends and title and label all figures and legends appropriately within the text.

Answer 23:

We have provided title, description and legend for the table and added figure captions and legends as needed within the text of the manuscript. 

Question 24:

The authors should also provide a legend or index for the supplementary information.

Answer 24:

We have added the captions of figures and tables to the Supplementary Information at the end of the article. We have separately uploaded supplementary information files.

Question 25:

Lines 190-191, please refrain from making statements commenting on the positivity or negativity of associations for statistically non-significant outcomes.

Answer 25:

We removed statements indicating statistically non-significant outcomes.

Question 26:

In the discussion section, please highlight the novelty of your findings. Were the results of the analysis unique and were they different from previous such analyses? If so, please provide a valid reason for this change in results. If not, please highlight the additional utility your findings may provide, either in the form of additional validation or simply being more up to date with current literature.

Answer 26:

We have highlighted the changes and extensions of the previous meta-analysis results in the Discussion section. 

(Line 332 in Discussion: Despite the current clinical literature examining the association between abdominal obesity and DR, conflicting results have been reported. Our results for the WC subgroup concur with those of the most recent meta-analyses, which demonstrated that abdominal obesity defined by WC is associated with the risk of DR. However, our results update their findings regarding WHR, for which the former study did not find a statistically significant association.

Line 408 in Strengths and limitations: First, our work is the first to systematically evaluate and meta-analyze the association between various parameters of abdominal obesity and DR in diabetic patients worldwide. Second, compared with previous meta-analyses among Chinese populations evaluated by WC and WHR, we included a broader population and more evaluation indicators. Finally, we assessed the effects of different parameters, DR severity, and ethnicity on outcomes using subgroup analysis and explored potential sources of heterogeneity.)

Question 27:

In the discussion section the authors should discuss the significance of their findings. How were these findings important? What do they demonstrate that may alter guidelines or inspire change in clinical or research practices in this field. The authors should also specifically mention any new avenues of research that have opened up as a result of this study.

Answer 27:

1.We have added the research significance of the article to the Supplementary Guidelines and guiding disease screening. 

(Line 318 in Discussion: The American Academy of Ophthalmology guidelines[33] conservatively mention a trend for stepwise increases in DR, corresponding to the number of MS components. Our study further clarified that abdominal obesity is independently associated with DR. The implications of these findings are quite important, since they may help illustrate that reducing abdominal obesity has a positive impact on the prevention of DR in patients with DM, regardless of ethnicity.

Line 336 in Discussion: Therefore, screening programs for WC and WHR may be more helpful in identifying patients at higher risk for DR progression than VFA, VAI, and LAP. Furthermore, considering economic expenditure, direct measurements of WC and WHR are more suitable for widespread screening.)

2.Meanwhile, we have added new avenues of research that have been opened up as a result of this study.

(Line 300 in Discussion: Whether these differences in ethnicity are exactly related to differences in abdominal fat accumulation is a question that remains to be answered by future research.

Line 405 in Discussion: These hypotheses on the mechanism of abdominal obesity require more in-depth clinical and basic verification.)

Question 28:

The article contains many grammatical and language errors. While the most significant ones have been highlighted above, many still remain which make the article confusing for the reader. The authors should revise the writing and should take care to amend these grammatical and language errors for the sake of readability and comprehension.

Answer 28:

We have comprehensively corrected the language and grammatical errors in the manuscript.

Thank you for your consideration. I look forward to hearing from you.

Sincerely,

Ximing Liu

---

## [Decision Letter · Decision Letter 1]

14 Dec 2022

Association between abdominal obesity and diabetic retinopathy in patients with diabetes mellitus: a systematic review and meta-analysis

PONE-D-22-22136R1

Dear Dr. Liu,

We’re pleased to inform you that your manuscript has been judged scientifically suitable for publication and will be formally accepted for publication once it meets all outstanding technical requirements.

Kind regards,

Tariq Jamal Siddiqi

Academic Editor

PLOS ONE

Additional Editor Comments (optional):

Reviewers' comments:

Reviewer's Responses to Questions

**Comments to the Author**

1. If the authors have adequately addressed your comments raised in a previous round of review and you feel that this manuscript is now acceptable for publication, you may indicate that here to bypass the “Comments to the Author” section, enter your conflict of interest statement in the “Confidential to Editor” section, and submit your "Accept" recommendation.

Reviewer #1: All comments have been addressed

Reviewer #2: All comments have been addressed

2. Is the manuscript technically sound, and do the data support the conclusions?

Reviewer #1: Yes

Reviewer #2: Yes

3. Has the statistical analysis been performed appropriately and rigorously? 

Reviewer #1: Yes

Reviewer #2: Yes

4. Have the authors made all data underlying the findings in their manuscript fully available?

Reviewer #1: Yes

Reviewer #2: Yes

5. Is the manuscript presented in an intelligible fashion and written in standard English?

Reviewer #1: Yes

Reviewer #2: Yes

6. Review Comments to the Author

Reviewer #1: (No Response)

Reviewer #2: (No Response)

7. PLOS authors have the option to publish the peer review history of their article (what does this mean?). If published, this will include your full peer review and any attached files.

Reviewer #1: **Yes: **Muhammad Talha Maniya

Reviewer #2: No

---

## [Editor Report · Acceptance letter]

21 Dec 2022

PONE-D-22-22136R1 

Association between abdominal obesity and diabetic retinopathy in patients with diabetes mellitus: a systematic review and meta-analysis 

Dear Dr. Liu:

I'm pleased to inform you that your manuscript has been deemed suitable for publication in PLOS ONE. Congratulations! Your manuscript is now with our production department. 

Kind regards, 

on behalf of

Dr. Tariq Jamal Siddiqi 

Academic Editor

PLOS ONE